# Mental health literacy in Arab states of the Gulf Cooperation Council: A systematic review

**Rowaida Elyamani** [1]*, **Sarah Naja**[1], **Ayman Al-Dahshan**[1]*, **Hamed Hamoud**[1],
**Mohammed Iheb Bougmiza**[2], **Noora Alkubaisi**[2]

1 Community Medicine Residency Program, Department of Medical Education, Hamad Medical Corporation, Doha, Qatar, 2 Community Medicine Residency Program, Department of Workforce Training, Primary Health Care Corporation, Doha, Qatar

☯ These authors contributed equally to this work.
* waida.elyamani@gmail.com (RE); ayman.aldahshan@hotmail.com (AAD)

## Abstract

### Background

Mental health literacy (MHL) has been relatively neglected, despite the increase of mental health illnesses worldwide, as well as within the Middle East region. A low level of MHL may hinder public acceptance of evidence-based mental health care.

### Aim

This systematic review aims to identify and appraise existing research, focusing on MHL among adults in the Gulf Cooperation Council (GCC) countries.

### Methods

A systematic search of electronic databases (PubMed, PsychInfo, and Medline) was carried out from database inception to July 2019, in order to identify peer-reviewed journal articles that investigated MHL in the GCC countries. Studies were eligible for inclusion if they were: cross-sectional studies, reported in English, targeted adults (aged 18 and above), conducted in any of the GCC countries, include at least one outcome measure of the main components of MHL: knowledge of mental illnesses and their treatment, stigmatizing attitudes towards mental illnesses, and seeking help for self and offering help.

### Results

A total of 27 studies (16,391 participants) were included. The outcome across studies varied due to disparity in the tested populations. Findings show that limited MHL was observed among participants, even health care professionals. Results also show a high cumulative level of stigma and negative attitude towards mental health illness in the public. Negative beliefs and inappropriate practices are common, as well. The majority of studies yielded a moderate to high risk of bias.

**Data Availability Statement:** All relevant data are within the manuscript and its Supporting information files.

**Funding:** This study is funded by Qatar National Library.

**Competing interests:** The authors have declared that no competing interests exist.

## Conclusion

This work indicates that research on MHL must be tackled through well-designed large-scale studies of the public. Campaigns to promote early identification and treatment of mental illness is also encouraged to improve overall level of MHL in the general population of the GCC region.

**Registration number**: PROSPERO 2018 CRD42018104492.

## Introduction

Mental health literacy (MHL) was first defined by Jorm AF as "knowledge and beliefs about mental disorders which aid their recognition, management, or prevention" [1]. Through the past decade, the concept evolved to include the importance of the ability to provide support to someone presenting with a mental health problem; that is, first aid skills [2].

MHL is a crucial element for promoting the mental health and well-being of populations overall. This is of great importance if we aim to overcome barriers of mental health, such as lack of knowledge, presence of stigma, and limited access to mental health care [3]. There is a large body of evidence emphasizing the positive association between adverse health outcomes and low MHL [4]; these problems are considered a global public health challenge, and are more common in young adults vs. other age groups. Such a challenge could be tackled early through the creation of a community with a high level of MHL [5, 6].

A systematic review on MHL in 2014 among eight Sub-Saharan African countries revealed that the number of available studies was scarce for scope, number, and spread. In the study, authors reported numerous limitations to existing studies and found that the majority of participants were unable to identify mental illnesses accurately, along with the unfavorable effect of sociocultural boundaries in their communities [7]. Another systematic review of MHL in non-Western countries showed overall adequate levels. However, when focusing on certain mental illnesses, such as anxiety and personality disorders, the level of MHL was generally low [8]. Since there is limited research that reflects MHL in the Middle East region, the picture still needs some clarity.

Gulf Cooperation Council (GCC) states, part of the Eastern Mediterranean region, share many social, religious, cultural, and economic features. Additionally, they share many of the same health challenges and opportunities. The GCC is comprised of six countries: Qatar, Saudi Arabia, United Arab Emirates (UAE), Oman, Kuwait, and Bahrain [9]. There are currently many papers published in the field of MHL, but to our knowledge, there are no reports of systematic reviews conducted to assess MHL in the GCC countries. Therefore, this review aims to explore MHL in the GCC countries as well as to uncover similarities, differences, and methodological issues among published studies.

## Materials and methods

A systematic review was conducted following the Preferred Reporting Items for Systematic Reviews and Meta-Analyses (PRISMA) guidelines [10].

### Search strategy for identification of studies

Three authors (R.E., S.N., and A.A.) independently performed a literature search in two electronic databases: PubMed and PsycINFO. Two authors (R.E. and H.H) explored the Medline

database for studies on MHL that had been published in any journal through July 2019 (without restriction to year of publication). A Boolean/phrase search was performed on each database, with search terms on the main concepts of interest: health literacy (concept 1), mental health (concept 2), and GCC countries (concept 3). (Details of the search strategy are included in the S1 Appendix).

## Inclusion and exclusion criteria

Inclusion criteria were: (a) cross-sectional studies with no time restrictions, (b) written in English, (c) age 18 and above, (d) conducted in any of the GCC countries, and (e) at least one outcome measure of the main components of MHL: knowledge of mental illnesses and their treatment, stigmatizing attitudes towards mental illnesses, and seeking help for self and offering help.

## Study selection

Four authors R.E., S.N., A.A., and H.H. independently screened titles and abstracts, and excluded studies that were not relevant to the topic. They reviewed the full-texts of articles. First, database searches were exported into a master folder. All titles and abstracts were screened by R.E. and H.H. and then screened by S.N. or A.A. to assess eligibility for full-text printing and screening of references. Further, these authors independently screened all excluded titles and abstracts. If there was a disagreement, it was discussed with M.B. or N.E. to reach a final decision.

## Data extraction

Independent data extraction of studies was performed by all four authors (R.E., S.N. A.A. and H.H.), to compare data and reach consensus. The following were extracted from each one: country, title, authors, time of study, design, population group, sample size, outcome measures, and the Newcastle-Ottawa Scale (NOS) score of the study. M.B. reviewed and adjusted the tables.

## Critical appraisal method

The quality of the studies and related bias were assessed by using NOS, adapted for cross-sectional studies [11]. This tool evaluated three quality parameters (selection, comparability, and outcome), divided across eight specific items. Each item on the scale was scored from one point, except for comparability, which can be adapted to the specific topic of interest to score up to two points. Thus, the maximum for each study is eight, with studies having less than four points identified as representing low quality. In order to minimize a subjective interpretation of bias in scoring the NOS, two independent authors should typically have scored each paper. All studies were assessed for quality in three domains: study selection, comparability, and outcome, with two authors (R.E. and H.H.) independently scoring the domains. When independent evaluations of the ranks differed between authors, they discussed the issues with a third author (A.A or S.N) to reach consensus.

Additionally, for the results synthesis in this systematic review, we have critically reviewed and scrutinized the results and discussion of each study for outcomes, limitations and bias that authors may have had highlighted in their studies. Following that we planned to combine and examine various related ideas in literature, to show how included results, outcomes, and limitations fit together, and present them in a unified form. Finally, we used the items from CASP

tool to draw limitations that were faced in this review, and further elaborate on them in the discussion.

## Outcomes

Studies must include at least one outcome measure, which was categorized as mental health attitudes (i.e., stigma, prejudice), knowledge of mental health (i.e., disorder and symptom recognition), or behavior regarding mental health (i.e., intended or actual help-seeking).

## Results

Fig 1 is the flow chart showing the procedure for selection of studies. We identified 341 studies in the initial search of databases. Next, we screened titles and ended with a total of 47. After removing 11 duplicates, we examined the abstracts of 36 potentially eligible studies, with nine of them excluded for not meeting the criteria for selection after fully reading the texts (three studies) or inaccessibility to the full text despite contacting the authors directly via email (six studies). Reference lists of these studies were screened as well, and finally 27 studies were included in this review.

## Characteristics of the studies and participants

Table 1 summarizes characteristics of reviewed studies: country, authors, study design, population, sample size, setting of study, measurement tools, and outcome. Overall, a total of 27 studies were included in the current systematic review, all of which were published after 2002. Regarding the setting, almost half the studies (13) were conducted in Saudi Arabia, six were conducted in the UAE, and four were conducted in Qatar (4). The remaining studies were conducted in Oman (2) and Kuwait (2). No studies could be found in Bahrain.

The included studies involved 16,391 participants who were primarily adults from the community. In six studies, participants were healthcare professionals [12, 19, 24, 26, 27, 37], and five studies were conducted among college students [16, 23, 25, 28, 31, 36]. In general, females constituted more than half of all study participants. Moreover, the majority of studies stated the definition of at least one component of MHL and explored the possible sociodemographic factors believed to influence the level of mental health knowledge, attitude, stigma, and MHL overall; specifically, gender [22, 31, 35], marital status, level of education [21], ethnic groups [33].

Common demographic predictors of a lower level in recognizing mental illnesses across studies were younger age, unemployment, illiteracy, and female gender. To the contrary, one paper from Saudi Arabia showed that gender and type of education (medical and non-medical students) were not significantly associated with the level of MHL. However, such findings cannot be generalized, as they included students from only one university [23].

## Level of mental health literacy

Three terms: knowledge, attitude, and practice were utilized in KSA and Qatar studies. The outcomes were similar, and a low level of knowledge, attitude, and practice were seen in KSA [20, 21], UAE [27] and Qatar [33, 34]. However, health professionals and students showed an intermediate level of knowledge in KSA [22, 23] and UAE [26]. Few publications reported only the attitude towards mental health, despite that results were similar across the studies, which agreed on the high level of stigma and shame [26, 31, 38]. However, none were conducted to explore self-efficacy and MHL in the GCC region.

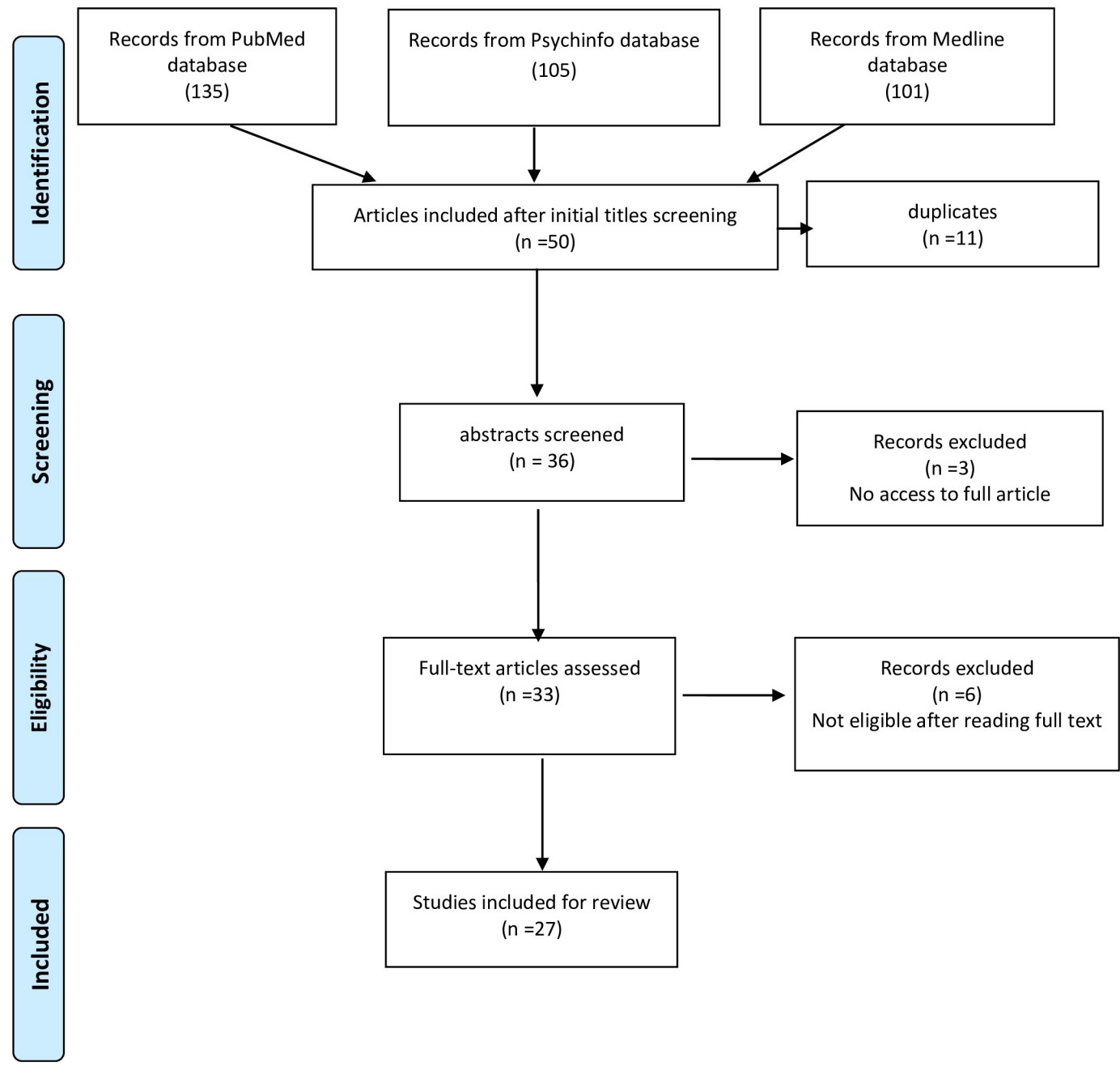

**Fig 1. Flow diagram.**

Almost all included studies in this review revealed an average level of MHL, with some studies showing lower levels than 50% of participants who could not recognize some common mental disorders. Also, the majority of papers revealed a high proportion of participants had negative attitudes toward mental disorders and people with mental illnesses. Some scholars focused on the identification of mental health disorders in general, their risk factors, common symptoms, and treatments or intervention options [22, 23, 31–38]. Others selected specific

**Table 1. Characteristics of the studies included in the systematic review.**

| Authors | Country | Population | Study design | Sample size | Settings | Study tools | Knowledge | Stigma | Self-efficacy | Study outcome |
|---|---|---|---|---|---|---|---|---|---|---|
| Aldahmashi T, Almanea A, Alsaad A, Mohamud M, Anjum I, 2019 [12] | Saudi Arabia | Non-psychiatric physicians | A cross-sectional study | 380 | Four government tertiary hospitals in Riyadh | Self-administered survey Validated and reliable tool | No | Yes | No | Overall, respondents were optimistic and had a positive perspective towards depression; results also showed that male physicians were more confident in depression care. |
| Abolfotouh MA, Almutairi AF, Almutairi Z, Salam M, Alhashem A, Adlan AA, Modayfer O, 2019 [13] | Saudi Arabia | Adults | A cross-sectional study | 642 | Saudi, annual cultural and heritage festival in Jenadriyah | Interview-based questionnaire Validated and reliable tool | Yes | Yes | No | Most participants had limited knowledge about the nature of mental illnesses. While more than half expressed negative attitudes towards mental diseases. |
| Aljedaani SM, 2018 [14] | Saudi Arabia | Adults | A cross-sectional study | 470 | Traditional and modern stores in Jeddah city | Self-administered survey Validated and reliable tool | Yes | No | No | Large numbers of responders held false believes related to causes of mental illness. |
| AlAteeq D, AlDaoud A, AlHadi A, AlKhalaf H, Milev R, 2018 [15] | Saudi Arabia | Adults with mood disorders | A cross-sectional study | 93 | Outpatient psychiatric clinic and psychiatric inpatient ward at King Saud University Medical City, Riyadh | Interview-based questionnaire Validated and reliable tool | No | Yes | No | More than half of participants reported trying to hide their mental illness in situations that might be stigmatizing. Almost half of participants with bipolar disorders and 42% with depression believed that the average person is afraid of a patient with a serious mental disease. |
| Alahmed S, Anjum I, Masuadi E, 2018 [16] | Saudi Arabia | Undergraduate health professional Students | A cross-sectional study | 233 | King Saud bin Abdulaziz University for Health Sciences, Riyadh | A self-administered questionnaire | Yes | Yes | No | Students mostly showed below average level of knowledge related to causes of mental illnesses. Additionally, the preferred solutions for mental issues were consultations and religious rituals. |
| Algahtani H, Shirah B, Alhazmi A, Alshareef A, Bajunaid M, Samman A, 2018 [17] | Saudi Arabia | Public adults | A cross-sectional study | 1698 | Malls and public places in Jeddah | A self-administered questionnaire | Yes | No | No | In general, findings revealed poor knowledge on mental illnesses, however there was much less stigmatizing attitudes and believes toward patients with mental problems. |

*(Continued)*

**Table 1.** (*Continued*)

| Authors | Country | Population | Study design | Sample size | Settings | Study tools | Knowledge | Stigma | Self-efficacy | Study outcome |
|---|---|---|---|---|---|---|---|---|---|---|
| Almutairi AF, Salam M, Alanazi S, Alweldawi M, Alsomali N, Alotaibi N. 2017 [18] | Saudi Arabia | Adult women | A cross-sectional study | 113 | Primary health care clinics in Riyadh city | Self-administered survey (Interview-based for illiterate) | No | No | Yes | The majority of respondents reported neutral help-seeking behavior. Among the most common sources of help were friends and family members. |
| | | | | | | Validated and reliable tools (GHSQ and PSS) | | | | |
| Al-Atram AA. 2018 [19] | Saudi Arabia | Physicians | A cross-sectional study | 142 | Hospitals | Self-administered survey | Yes | No | No | GPs had better knowledge about depression than anxiety in contrary to family practitioners, while specialist's knowledge of both disorders were 74% and 63% respectively. |
| Alosaimi FD, AlAteeq DA, Bin Hussain SI, Alhenaki RS, Bin Salamah AA, AlModihesh NA. 2019 [20] | Saudi Arabia | Adults | A cross-sectional study | 186 | Malls, university, hospitals | Interview-based questionnaire | Yes | Yes | No | 48% of participants lacked knowledge about bipolar disorder. |
| | | | | | | Validated and reliable tool | | | | Attitude was negative and stigmatizing. |
| | | | | | | | | | | 50% don't believe in medication treatment for mental health disorders and they prefer religious strategies to heal. |
| Siddiqui A, Mahasin S, Alsajjan R, Hassounah M, Alhalees Z, AlSaif N et al. 2017 [21] | Saudi Arabia | Adult women | A cross-sectional study | 409 | Outpatient clinics at King Khalid University Hospital | Self-administered survey | No | No | No | Low health literacy for depression in 35%. With mostly negative attitude toward people with mental illnesses. |
| | | | | | | Validated and reliable tool | | | | |
| Khalil A. 2017 [22] | Saudi Arabia | Adults | A cross-sectional study | 255 | Shopping malls, universities, and restaurants | Interview-based questionnaire &self-administrated | Yes | Yes | No | Adequate level of knowledge about causes and treatment for mental illnesses with significant difference between men and women, 57% indicted negative attitude and stigmatization. |
| | | | | | | Validated tool | | | | |
| Mahfouz, M. S., Aqeeli, A., Makeen, A. M., Hakami, R. M., Najmi, H. H., Mobarki, A. T., 2016 [23] | Saudi Arabia | University students (18 to 28 years) | A cross-sectional study | 557 | Jazan University | Interview-based questionnaire | Yes | Yes | No | The majority of students have intermediate mental health literacy. They found that stigma mainly causes poor social relationships. |
| | | | | | | Validated and reliable tool | | | | |

(*Continued*)

**Table 1.** (Continued)

| Authors | Country | Population | Study design | Sample size | Settings | Study tools | Knowledge | Stigma | Self-efficacy | Study outcome |
|---------|---------|-----------|--------------|-------------|----------|-------------|-----------|--------|---------------|---------------|
| Mohammed N. Al-Arifi [24] | Saudi Arabia | Community pharmacists | A cross-sectional study | 43 | College of Pharmacy, King Saud University, Riyadh, | A self-administered questionnaire | No | Yes | No | Overall, pharmacists expressed positive attitudes toward mental illness and the provision of pharmaceutical care to mentally-ill patients, however, they reported feeling more uncomfortable counseling, and solving drug-related problems for those patients. |
| Vally Z, Brettjet L, Cody, Maryam A. Albloshi, Safeya N. M. Alsheraifi [25] | UAE | Female undergraduate students | A cross-sectional study | 114 | Undergraduate at a federal university | Online survey Validated tools | No | Yes | No | The study participants showed high levels of both public stigma and self-stigma. However, psychology students showed lower levels of stigma as well as more favorable attitudes toward seeking psychological help. |
| Al-Yateem N, Rossiter R, Robb W, Slewa-Younan, S. 2018 [26] | UAE | Schools nurses | A cross-sectional study | 324 | Schools | Interview-based questionnaire Validated and reliable tool | Yes | No | No | Less than 50% of the respondents correctly identified the disorders presented, while accurate identification of evidence-based interventions was also limited. |
| Al-Yateem N, Rossiter R, Robb W, et al. 2017 [27] | UAE | Healthcare professionals | A cross-sectional study | 317 | Hospitals | Interview-based questionnaire | Yes | No | No | Correct identification of the diagnosis for posttraumatic stress disorder, depression with suicidal thoughts limited recognition of mental health disorders, ranging from 47% for PTSD to 54.3% for psychosis and recognition of intervention was about 50%. |
| Al-Darmaki F, Thomas J, Yaaqeib, S. 2015 [28] | UAE | University students Age (18–42) | A cross-sectional study | 70 | At university | Self-administered questionnaire | Yes | Yes | No | Majority lacked adequate knowledge of psychological disorders and held false believes about causes of mental illness. |

(*Continued*)

**Table 1.** (Continued)

| Authors | Country | Population | Study design | Sample size | Settings | Study tools | Knowledge | Stigma | Self-efficacy | Study outcome |
|---|---|---|---|---|---|---|---|---|---|---|
| Salem MO, Saleh B, Yousef S, Sabri S. [29] | UAE | Adult psychiatric patients | A cross-sectional study | 106 | Al Ain Hospital (outpatient and inpatient) | Interview-based questionnaire | Yes | Yes | No | About half of respondents consulted either faith healers or their primary care physician before presenting to the secondary psychiatric care. Only one-third believed to have a psychiatric illness. |
| Eapen V, Ghubash R. 2004 [30] | UAE | Parents | A cross-sectional study | 325 | Households in Al Ain, UAE | Semi structured interview schedule | No | Yes | Yes | Main reasons given for not seeking a professional advice were reluctance to acknowledge that a member of their family has a mental illness, stigma attached to attending mental health services, and the doubt about the effectiveness of mental health services. |
| Zolezzi M, Bensmail N, Zahrah F, Khaled SM, El-Gaili T 2017 [31] | Qatar | University students | A cross-sectional study | 282 | Universities | Interview-based questionnaire | No | Yes | No | Majority of the students reported stigmatizing believes and attitude. |
| Ghuloum, S & Bener, A. 2011 [32] | Qatar | Arab adult population above 20 years of age. | A cross-sectional study | 2,514 | Primary health care centers | Interview-based questionnaire Validated and reliable tool | Yes | Yes | No | Wrong perceptions about mental illnesses were very common. Qatari citizens had a poor knowledge about causes of mental illness compared to non-Qatari Arabs. |
| Ghuloum, S & Bener, A. 2010 [33] | Qatar | General public | A cross-sectional study | 2,514 | Primary health care centers | Interview-based questionnaire Validated and reliable tool | Yes | Yes | No | Gender difference in knowledge, attitudes and practice towards mental illness. Beliefs of evil spirit and traditional healers were more among women than men. Men had a better knowledge, beliefs and attitude towards mental illness than women. |

(*Continued*)

**Table 1.** (Continued)

| Authors | Country | Population | Study design | Sample size | Settings | Study tools | Knowledge | Stigma | Self-efficacy | Study outcome |
|---|---|---|---|---|---|---|---|---|---|---|
| Bene A, Ghuloum S. 2010 [34] | Qatar | General public | A cross-sectional study | 2254 | Primary health care centers | Interview-based questionnaire  Reliable and valid tool | Yes | Yes | No | Poor knowledge was identified among the majority, and the most common source of mental health knowledge was media. |
| Alawi M, Sinawi H, AL-Adawi S, Jeyaseelan L, MurthiP S 2017 [35] | Oman | Public | A cross-sectional study | 601 | Online survey | Online survey  Perception and attitude  Validated and reliable tool | Yes | No | No | Findings showed high level of literacy |
| Al-Adawi S, Dorvlo A S S, Al-Ismaily S S, Al-Ghafry D A, Al-Noobi B Z, Al-Salmi A, et al. 2002 [36] | Oman | Medical students, relatives of people with mental illness, and the general Omani public | A cross-sectional study | 458 | University, outpatient psychiatric clinic, community | Interview-based questionnaire | No | Yes | No | Omani physicians showed negative attitude toward mental health. |
| Al-Awadhi A, Atawneh F, Alalyan MY, Shahid AA, Al-Alkhadhari S, Zahid MA. 2017 [37] | Kuwait | nurses | A cross-sectional study | 990 | Hospital | Self-administered survey  CAMI Scale | No | Yes | No | Nurses' attitudes toward mental illness were generally negative. |
| Meguid M A, Rabie M A, Bassim R E. 2010 [38] | Kuwait | Non-medical staff in psychiatric hospitals | A cross-sectional study | 301 | Psychiatric hospitals | Interview-based questionnaire  CAMI Scale | No | Yes | No | The study showed stigma in two different countries. Knowledge and practice were not studied. |

mental health illnesses, such as depression [21, 26, 27] bipolar illness [20], post-traumatic stress disorder (PTSD), or psychosis [26, 27].

Researchers in the UAE focused primarily on MHL among healthcare professionals and students. Results of the study on pediatric department medical staff revealed an average level of MHL and limited recognition of common mental disorders. Additionally, there was a significant association between any form of psychosocial distress and choosing the correct depression diagnosis (P = 0.01) [26]. In a second study of nurses, less than half of respondents accurately identified selected mental disorders in specific cases and their appropriate evidence-based interventions; data showed no significant association with problem recognition and beliefs about interventions [27], while nurses in a Kuwaiti study conveyed a significantly negative attitude toward mental illness [37, 38]. Healthcare participants from Saudi Arabia demonstrated a high level of knowledge on anxiety and depression, especially among family practitioners and specialists [19].

## Quality of studies

Table 2 shows the scoring of studies using the NOS tool. First, selection of adult participants in reviewed studies was from different populations, but not usually justified. Most studies reported specific inclusion and exclusion criteria. Probability sampling was conducted in few

**Table 2. Scoring of reviewed studies using Newcastle-Ottawa Scale.**

| Study | Representativeness of the sample | Sample size | Non-respondents | Risk factor assessment | Control of confounders (Up to 2 stars) | Outcome assessment | Statistical test | Score (0–10) |
|---|---|---|---|---|---|---|---|---|
| | a) Truly representative of the average in the target population. * b) Somewhat representative of the average in the target population. * c) Selected group of users. d) No description of the sampling strategy. | a) Justified and satisfactory. * b) Not justified. | a) The response rate is satisfactory. * b) The response rate is unsatisfactory c) No description of the response rate | a) Validated measurement tool.** b) Non-validated measurement tool, but the tool is available or described. * c) No description of the measurement tool. | a) The study controls for the most important factor (select one). * b) The study control for any additional factor. * | a) Validated measurement tool. ** b) Non-validated measurement tool, but the tool is available or described. ** c) Self-reporting outcome. * d) No description of the measurement tool. | a) The statistical test used to analyze the data is clearly described and appropriate. * b) The statistical test is not appropriate, not described or incomplete. | |
| Aldahmashi T, Almanea A, Alsaad A, Mohamud M, Anjum I, 2019 [12] | No | Yes* | No | Yes* | Yes* | Yes** | Yes* | 6 |
| Abolfotouh MA, Almutairi AF, Almutairi Z, Salam M, Alhashem A, Adlan AA, Modayfer O, 2019 [13] | Yes* | No | No | Yes* | Yes* | Yes** | Yes* | 6 |
| Aljedaani SM, 2018 [14] | No | Yes* | No | Yes* | No | Yes** | Yes* | 5 |
| AlAteeq D, AlDaoud A, AlHadi A, AlKhalaf H, Milev R,2018 [15] | No | No | Yes* | Yes* | Yes* | Yes** | Yes* | 6 |
| Alahmed S, Anjum I, Masuadi E, 2018 [16] | Yes* | Yes | Yes* | No | Yes* | Yes** | Yes* | 7 |
| Algahtani H, Shirah B, Alhazmi A, Alshareef A, Bajunaid M, Samman A, 2018 [17] | Yes* | Yes | No | No | No | Yes** | Yes* | 5 |
| Almutairi AF, Salam M, Alanazi S, Alweldawi M, Alsomali N, Alotaibi N. 2017 [18] | Yes* | No | Yes | Yes* | Yes* | Yes** | Yes* | 7 |
| Al-Atram AA. 2018 [19] | Yes* | No | Yes* | Yes* | Yes* | Yes** | Yes* | 7 |
| Alosaimi FD, AlAteeq DA, Bin Hussain SI, Alhenaki RS, Bin Salamah AA, AlModihesh NA. 2019 [20] | Yes* | No | No | Yes* | No | Yes** | Yes* | 5 |

*(Continued)*

**Table 2.** (Continued)

| | | | | | | | | |
|---|---|---|---|---|---|---|---|---|
| Siddiqui A, Mahasin S, Alsajjan R, Hassounah M, Alhalees Z, AlSaif N et al. 2017 [21] | Yes* | No | Yes* | Yes* | No | Yes** | Yes* | 6 |
| Khalil A. 2017 [22] | No | No | Yes* | Yes* | No | Yes** | Yes | 5 |
| Mahfouz, M. S., Aqeeli, A., Makeen, A. M., Hakami, R. M., Najmi, H. H., Mobarki, A. T., 2016 [23] | Yes* | Yes* | Yes* | Yes | Yes* | Yes** | Yes* | 7 |
| Mohammed N. Al-Arifi [24] | No | No | Yes* | Yes* | Yes* | Yes* | No | 4 |
| Vally Z, Brettjet L, Cody, Maryam A. Albloshi, Safeya N. M. Alsheraifi [25] | Yes* | Yes* | No | Yes | Yes** | Yes** | Yes* | 8 |
| Al-Yateem N, Rossiter R, Robb W, Slewa-Younan, S. 2018 [26] | Yes* | No | No | Yes** | Yes** | Yes* | Yes* | 7 |
| Al-Yateem N, Rossiter R, Robb W, et al. 2017 [27] | Yes* | Yes* | No | Yes** | Yes** | Yes* | Yes* | 8 |
| Al-Darmaki F, Thomas J, Yaaqeib, S. 2015 [28] | No | No | No | Yes* | Yes* | Yes** | Yes* | 5 |
| Salem MO, Saleh B, Yousef S, Sabri S. [29] | No | No | Yes* | Yes* | Yes* | Yes* | No | 4 |
| Eapen V, Ghubash R. 2004 [30] | No | No | Yes* | Yes* | Yes* | Yes* | Yes* | 5 |
| Zolezzi M, Bensmail N, Zahrah F, Khaled SM, El-Gaili T 2017 [31] | No | No | Yes* | Yes* | No | Yes* | Yes* | 4 |
| Ghuloum, S & Bener, A. 2011 [32] | Yes* | No | Yes* | Yes* | Yes* | Yes** | Yes* | 7 |
| Ghuloum, S & Bener, A. 2010 [33] | Yes* | No | Yes* | Yes* | Yes* | Yes** | Yes* | 7 |
| Bene A, Ghuloum S. 2010 [34] | Yes* | No | Yes* | Yes* | Yes* | Yes** | Yes* | 7 |
| Alawi M, Sinawi H, AL-Adawi S, Jeyaseelan L, MurthiP S 2017 [35] | No | No | Yes* | Yes* | No | Yes** | Yes* | 5 |
| Al-Adawi S, Dorvlo A S S, Al-Ismaily S S, Al-Ghafry D A, Al-Noobi B Z, Al-Salmi A, et al. 2002 [36] | No | Yes* | No | Yes* | Yes* | Yes* | Yes* | 5 |

(*Continued*)

**Table 2.** (Continued)

| | | | | | | | | |
|---|---|---|---|---|---|---|---|---|
| Al-Awadhi A, Atawneh F, Alalyan MY, Shahid AA, Al-Alkhadhari S, Zahid MA. 2017 [37] | Yes* | Yes* | No | Yes** | Yes* | Yes** | Yes* | 8 |
| Meguid M A, Rabie M A, Bassim R E. 2010 [38] | Yes* | Yes* | Yes* | Yes* | Yes* | Yes* | Yes* | 8 |

studies, which renders results representative of the selected population, such as random sampling [38], cluster probability [26, 27], and multistage stratified sampling design [12, 19, 32–34]. The remaining studies utilized nonprobability sampling from the community [13–15, 22–25, 28–31].

Moreover, a high level of non-response was noted in a few studies, such as 77% [26], 53% [27], and 69% [37], which could probably have been avoided if they analyzed the characteristics of non-respondents. Some studies where purely descriptive, with no comparison among different sub-groups, which compromised their data via the confounding effect [26]. Results were not subjected to multivariate analysis, such as structural equation modeling; they did not test causal theories of how individual difference factors or experiences may have influenced their results.

Studies were heterogeneous in terms of outcome measures. The vast majority of papers utilized valid and reliable measurement tools; these tools were piloted and tested within the populations, thereby making these studies less vulnerable to measurement bias. Some studies adapted validated tools such as community attitudes toward the mentally ill, the CAMI Scale [37, 38], while others used measurement instruments that fulfilled the Diagnostic and Statistical Manual of Mental Disorders, Fifth Edition (DSMV) [26, 27].

Furthermore, Quantifying the differences in means of levels of mental health literacy that were reported across included studies was not possible due to differences in measuring the outcomes and the scales used in addition to the selection of certain mental diseases to assess the level of mental health literacy related to them across different population groups. For example, adult females, health care providers or general adults.

## Discussion

This systematic review provides a narrative synthesis for MHL in the Arab Gulf countries. Most studies reported significantly low levels of MHL in the general public. Moreover, three methodological issues could compromise the validity and reliability of the results. First, different terminologies were used, such as health literacy, knowledge, attitude, practice, health-seeking behavior, and stigma. Second, many included studies used different methods to assess recognition of mental illness or the outcome overall, which can lead to measurement bias. Third, variations in the population under study including university students, adults, and healthcare providers, which prevented cross cultural comparisons between countries on these studies.

Studies generally yielded average quality, based on the NOS assessment scores; this is likely due to potential sources of bias, which may compromise internal validity. We are unable to draw robust conclusions from these articles. More papers were from Saudi Arabia and UAE vs. other countries, and they investigated heterogeneous populations, so findings may not be

generalizable to other contexts or populations. Despite these flaws, the included studies contain valuable information.

The discrepancies in terminology to define MHL, and the selection of different mental illnesses for included studies made it difficult to perform cross-country comparisons. This contradicted some findings from a review article, reporting on incomparable results despite using similar methodologies in countries from developed and developing regions: authors attributed the issues as being linked to religious beliefs, hindering interventions for populations with low MHL [39].

In the UAE, they used related case scenarios for recognition of PTSD, depression with suicidal thoughts and psychoses (Miriam, Abdul, and Saeed) [26, 27]. In Saudi Arabia, they asked general questions about knowledge, attitude, and health-seeking behaviors or beliefs [13–16, 22–24]. These findings were similar to those reported by Furnham and Swami in their review article, where case scenarios were commonly used in studies to assess MHL in the general public [39]. Regarding type of interviews, some vignettes were used in telephone interviews, but there were no video clips of "actual people" with mental disorders in the reviewed studies, which might be culturally unacceptable, and stigmatizing to both Arab and Islamic countries.

Other methods were used in included studies to assess the MHL level. For example, researchers from Saudi Arabia investigated causes, knowledge, attitudes, and management of mental disorders on a 5-point scale (strongly agree, agree, neutral, disagree, and strongly disagree) [23]. Compared to two studies in the UAE, categorized answers were used in seeking treatment as helpful, harmful, neither, or most helpful [26, 27]. Other studies used four Likert scores: 'strongly agree,' 'agree,' 'disagree,' and 'strongly disagree) [21], and two-Likert scores: (Yes, No) [22]. Differences in the cut-off points led to potential misclassification biases.

## Comparisons with previous evidence

Many studies suggest that a higher level of MHL is expected in the general population in developed countries, compared to their counterparts in developing regions. For instance, Furnham and Hamid in a systematic review (2014), examined a decade of research on MHL and found that citizens in developed countries had greater MHL: for example, the majority of reviewed studies showed that people had a greater recognition of depression than schizophrenia [8]. Authors justified their results in that the research from developing regions often used medical jargon and obscure language for the lay individual; in addition, restricted use of a written format for assessments rather than other forms of communications, such as audios and visuals; could complicate the situation [8]. Our study focused on the public, in which average MHL was found; these individuals were unable to recognize symptoms of common mental illnesses (e.g., depression and schizophrenia). A number of influencing factors could account for an opposing conclusion, such as literacy levels, cultural boundaries, and religious beliefs.

Gender differences varied across the literature; however, we found no significant difference between men and women in the GCC region in levels of MHL overall. To the contrary, Western studies show that women display better MHL than men [40, 41]. Other studies indicate that men are less able to correctly identify symptoms of mental illness in case vignettes, but are more likely to suggest self-help treatments [42–44]. Authors rationalize this by claiming that women favor psychological explanations for causes of mental illness, and are more open to psychological interventions, which is not the case in the Arab region, where mostly both men and women tend to link emotional and mental issues with religious and social believes.

Knowledge about mental health illness is lacking among the public as well as health care providers, in the available literature nowadays, despite the general perception that health care providers are more equipped to deal with patients suffering from mental illnesses. In this

review we found low levels of MHL among physicians, nurses, and other related health care workers. These findings are related to some studies that revealed limited knowledge and awareness of common mental disorders among health care providers, that coincides with false beliefs mostly acquired from their community and cultural environments [45, 46]. This might potentially create a real barrier in achieving accessible and effective mental health services to the public and especially to vulnerable groups within the GCC region who may struggle to reach out for their primary health provider voluntary.

Internationally, mental health interventions were used in health promotion models for individuals, schools, and communities [47]. A number of successful awareness campaigns that targeted the general public was reported in the UK, Norway, and the USA, in which media were used to reach larger audiences [48]. Additionally, an Australian trial set out to assess the effectiveness of an intervention called 'First Aid,' which showed remarkable results among adults in the workplace [49]. The use of such methods for improving MHL in developing countries requires further study. In this review, interventional studies were not identified in GCC countries, which can be explained by multilevel obstacles hindering strategy implementation, such as acceptability of interventions at the cultural level [50]. MHL levels in the GCC must be accelerated in the future.

This review was limited to peer-reviewed articles published in English, which could result in bias. Countries in the GCC uses Arabic as the first language spoken, however, English is the second Language spoken and the formal language used in medical field including both practice as well as academic and scientific professional research. There, is only a few numbers database that covers articles written originally in Arabic language, but they mostly cover only limited numbers of articles from wide range of topic areas such as art and engineering, but not exclusively on health nor mental health in specific. Hence choosing English was the best choice that would yield more studies in searching. While extensive search was conducted, it is possible that relevant articles were not identified, as authors were using specific scientific databases. The three databases selected were among the most commonly used in systematic reviews. Relating to similar literature on the topic from the region, these same databases were commonly and most frequently used. Never the less, including more data bases could have yield more articles, but it wasn't feasible nor convenient to add more search engine in the allocated time of the review. Authors also did not include information from other sources, such as unpublished reports from educational institutions or relevant literature. Our authors did not contact other authors to clarify vague information in reviewed studies. Thus, the evaluations may poorly assess the study quality, when details are not included in the reports.

## Conclusions

This review promotes common issues that shape and influence the level of MHL across GCC countries. These findings also suggest that there is a great need for interventions and public campaigns to both increase and promote MHL among the public. In addition, it emphasizes the need for robust cohort and interventional studies, given the importance of mental health, as well as its impact on the general well-being of the population.

## Supporting information

**S1 Appendix. Search strategy.**
(DOCX)

**S1 Checklist. PRISMA 2009 checklist.**
(DOC)

## Author Contributions

**Conceptualization:** Rowaida Elyamani, Mohammed Iheb Bougmiza, Noora Alkubaisi.

**Data curation:** Rowaida Elyamani, Sarah Naja, Ayman Al-Dahshan, Hamed Hamoud.

**Formal analysis:** Rowaida Elyamani, Sarah Naja, Hamed Hamoud.

**Investigation:** Ayman Al-Dahshan, Hamed Hamoud.

**Methodology:** Rowaida Elyamani, Sarah Naja, Noora Alkubaisi.

**Project administration:** Rowaida Elyamani.

**Supervision:** Mohammed Iheb Bougmiza, Noora Alkubaisi.

**Writing – original draft:** Rowaida Elyamani, Ayman Al-Dahshan, Hamed Hamoud, Mohammed Iheb Bougmiza.

**Writing – review & editing:** Rowaida Elyamani, Sarah Naja, Hamed Hamoud.

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
