## [Decision Letter · Decision Letter 0]

18 May 2020

PONE-D-20-06289

Mental health literacy in Arab states of the Gulf Cooperation Council: A systematic review

PLOS ONE

Dear Dr Elyamani,

Thank you for submitting your manuscript to PLOS ONE. After careful consideration, we feel that it has merit but does not fully meet PLOS ONE’s publication criteria as it currently stands. Therefore, we invite you to submit a revised version of the manuscript that addresses the points raised during the review process.

Your manuscript would be much improved if you follow the PRISMA checklist for systematic reviews reporting. I suggest you in particular rewrite your abstract adding details of your studies inclusion and exclusion criteria. In addition, I recommend you follow the indications of referees, namely referee #1 suggest to reflect on your conclusions and expand your discussion points; referee #2 invites you to indicate if there are available tools to promote mental health literacy. For example tools from other part of the world could be locally adapted.

In addition, I would suggest concluding with a few remarks of possible strategies to promote mental health literacy in the Arab States and beyond. 

We would appreciate receiving your revised manuscript by Jul 02 2020 11:59PM. To enhance the reproducibility of your results, we recommend that if applicable you deposit your laboratory protocols in protocols.io, where a protocol can be assigned its own identifier (DOI) such that it can be cited independently in the future. For instructions see: http://journals.plos.org/plosone/s/submission-guidelines#loc-laboratory-protocols

We look forward to receiving your revised manuscript.

Kind regards,

Marica Ferri

Academic Editor

PLOS ONE

Journal Requirements:

2. Please provide the full search string used, which at the moment is not included in the Supplementary file 1.

Reviewers' comments:

Reviewer's Responses to Questions

**Comments to the Author**

1. Is the manuscript technically sound, and do the data support the conclusions?

Reviewer #1: Yes

Reviewer #2: Yes

2. Has the statistical analysis been performed appropriately and rigorously? 

Reviewer #1: N/A

Reviewer #2: Yes

3. Have the authors made all data underlying the findings in their manuscript fully available?

Reviewer #1: Yes

Reviewer #2: Yes

4. Is the manuscript presented in an intelligible fashion and written in standard English?

Reviewer #1: No

Reviewer #2: Yes

5. Review Comments to the Author

Reviewer #1: This is a relatively well designed, run and written systematic review focusing on Mental health literacy (MHL) in Arab states of the Gulf. Authors identified a moderate number of studies meeting their inclusion criteria showing that limited MHL was observed among participants, even within health care professionals, high level of stigma and negative attitude toward

mental health illnesses among the public. Negative beliefs and inappropriate practices were common, as well. The majority of studies yielded moderate to high risk of bias. It is appreciable authors explored a topic which has local but also some general interest. Provided language will be edited by some native English speaker ( for example authors not reviewers perform screening ) it might deserve publication.

What would need significant additional thoughts and work is the discussion paragraph, as it's often the case for systematic review and meta-analysis. Authors should try and expand the meaning of their findings over and above the mere technical results. Comparisons with available literature and alternative explanations attempts are currently quite poor.

Reviewer #2: It is an important and topical issue.The review will help identify the gaps , however it will be helpful to identify some evidence informed interventions which could promote mental health literacy in specific population groups

6. PLOS authors have the option to publish the peer review history of their article (what does this mean?). If published, this will include your full peer review and any attached files.

Reviewer #1: No

Reviewer #2: No

---

## [Author Response · Author response to Decision Letter 0]

30 Jun 2020

General Response to the Reviewers’ Comments: ‎

We are appreciative of the insightful comments raised by the Academic Editor and the ‎reviewers. We believe we have responded to each of their comments. Where necessary, we ‎have made revisions in the manuscript. These revisions are tracked. The line numbers referred ‎to in our responses below refer to those in the tracked version of the manuscript. ‎

ACADEMIC EDITOR:‎

Your manuscript would be much improved if you follow the PRISMA checklist for systematic ‎reviews reporting. I suggest you in particular rewrite your abstract adding details of your ‎studies inclusion and exclusion criteria.‎

Response: Thank you for your insightful observation, we have followed the PRISMA checklist ‎and added more details in the abstract section. Please see lines 32-36.‎

REVIEWERS COMMENTS: 

Comment #1: ‎

Is the manuscript presented in an intelligible fashion and written in standard English?‎

Reviewer #1: No

Reviewer #2: Yes ‎

Response: ‎

Thank you for this feedback. We have engaged the services of a native English speaker to ‎‎review the entire manuscript and make the necessary corrections.‎

Comment #2: (Reviewer #1)‎

This is a relatively well designed, run and written systematic review focusing on Mental health ‎literacy (MHL) in Arab states of the Gulf. Authors identified a moderate number of studies ‎meeting their inclusion criteria showing that limited MHL was observed among participants, ‎even within health care professionals, high level of stigma and negative attitude toward 

mental health illnesses among the public. Negative beliefs and inappropriate practices were ‎common, as well. The majority of studies yielded moderate to high risk of bias. It is appreciable ‎authors explored a topic which has local but also some general interest. Provided language will ‎be edited by some native English speaker (for example authors not reviewers perform ‎screening) it might deserve publication.‎

What would need significant additional thoughts and work is the discussion paragraph, as it's ‎often the case for systematic review and meta-analysis. Authors should try and expand the ‎meaning of their findings over and above the mere technical results. Comparisons with ‎available literature and alternative explanations attempts are currently quite poor.‎

Response: ‎

Thank you for your insightful feedback. We have addressed your comments and the discussion ‎section has been modified accordingly. Kindly see the details highlighted below:‎

Lines 239-246 (discussion section)‎

The discrepancies in terminology to define MHL, and the selection of different mental illnesses ‎for included studies made it difficult to perform cross-country comparisons. This contradicted ‎some findings from a review article, reporting on incomparable results with similar ‎methodologies in countries of developed and developing regions: authors attributed the issues ‎as being linked to religious beliefs, hindering interventions for populations with low MHL [39]. ‎

Lines 279-294 (discussion section)‎

Many studies suggest that a higher level of MHL is expected in the general population in ‎developed countries, compared to their counterparts in developing regions. For instance, ‎Furnham and Hamid in a systematic review (2014), examined a decade of research on MHL and ‎found that citizens in developed countries had greater MHL: for example, the majority of ‎reviewed studies showed that people had a greater recognition of depression than ‎schizophrenia [8]. ‎

Authors justified their results in that the research from developing regions often used medical ‎jargon and obscure language for the lay individual; in addition, restricted use of a written ‎format for assessments rather than other forms of communications, such as audios and visuals, ‎could complicate the situation [8]. Our study focused on the public, in which average MHL was ‎found; these individuals were unable to recognize symptoms of common mental illnesses (e.g., ‎depression and schizophrenia). A number of influencing factors could account for an opposing ‎conclusion, such as literacy levels, cultural boundaries, and religious beliefs.‎

Lines 306-312 (discussion section)‎

Knowledge about mental health illness is lacking among the public as well as health care ‎providers, in the available literature nowadays, despite the general perception that health care ‎providers are more equipped to deal with patients suffering from mental illnesses. In this ‎review we found low levels of MHL among physicians, nurses, and other related health care ‎workers. These findings are related some studies that revealed limited knowledge and ‎awareness of common mental disorders among health care providers, that coincides with false ‎beliefs mostly acquired from their community and cultural environments [45,46]. ‎

Lines 313-325 (discussion section)‎

Internationally, mental health interventions were used in promotion models for individuals, ‎schools, and communities [47]. A number of successful awareness campaigns that targeted the ‎general public was reported in the UK, Norway, and the USA, in which media were used to ‎reach larger audiences [48]. Additionally, an Australian trial set out to assess the effectiveness ‎of an intervention called ‘First Aid,’ which showed remarkable results among adults in the ‎workplace [49]. The use of such methods for improving MHL in developing countries requires ‎further study. In this review, interventional studies were not identified in GCC countries, which ‎can be explained by multilevel obstacles hindering strategy implementation, such as ‎acceptability of interventions at the cultural level [50]. MHL levels in the GCC must be ‎accelerated in the future.‎

Lines 335-342 (conclusion section)‎

This review promotes common issues that shape and influence the level of MHL across GCC ‎countries. These findings also suggest that there is a great need for interventions and public ‎campaigns to both increase and promote MHL among the public. In addition, it emphasizes the ‎need for robust cohort and interventional studies, given the importance of mental health, as ‎well as its impact on the general well-being of the population. ‎

Comment #3: (Reviewer #2)‎

It is an important and topical issue. The review will help identify the gaps, however it will be ‎helpful to identify some evidence informed interventions which could promote mental health ‎literacy in specific population groups

Response: ‎

Thank you for your insightful comment. We have addressed your comment in the discussion ‎section. Kindly see the details highlighted below:‎

Lines 313-325 (discussion section)‎

Internationally, mental health interventions were used in promotion models for individuals, ‎schools, and communities [47]. A number of successful awareness campaigns that targeted the ‎general public was reported in the UK, Norway, and the USA, in which media were used to ‎reach larger audiences [48]. Additionally, an Australian trial set out to assess the effectiveness ‎of an intervention called ‘First Aid,’ which showed remarkable results among adults in the ‎workplace [49]. The use of such methods for improving MHL in developing countries requires ‎further study. In this review, interventional studies were not identified in GCC countries, which ‎can be explained by multilevel obstacles hindering strategy implementation, such as ‎acceptability of interventions at the cultural level [50]. MHL levels in the GCC must be ‎accelerated in the future.‎

Journal Requirements:‎

‎1- When submitting your revision, we need you to address these additional requirements: ‎Please ensure that your manuscript meets PLOS ONE's style requirements, including those for ‎file naming. ‎

Response: We have checked the style used in our manuscript against those required by PLOS ‎ONE.‎

‎2- Please provide the full search string used, which at the moment is not included in the ‎Supplementary file 1.‎

Response: We have provided the full search string used in the Supplementary file 1.‎

---

## [Decision Letter · Decision Letter 1]

20 Aug 2020

PONE-D-20-06289R1

Mental health literacy in Arab states of the Gulf Cooperation Council: A systematic review

PLOS ONE

Dear Dr. Elyamani,

Thank you for submitting your manuscript to PLOS ONE. After careful consideration, we feel that it has merit but does not fully meet PLOS ONE’s publication criteria as it currently stands. Therefore, we invite you to submit a revised version of the manuscript that addresses the points raised during the review process.

We suggest in particular you go through the PRISMA checklist providing explanations for all the items that were not met an include a synthesis in the limitations of the study. For example you may wish to expand on the reasons why you only included articles in English and you did not consider publications in Arab Languages, as suggested by Reviewer 3. You may elaborate also on the choice to include a selected number of articles databases and how this may have restricted the number of publications considered in your review.

We look forward to receiving your revised manuscript.

Kind regards,

Marica Ferri

Academic Editor

PLOS ONE

Reviewers' comments:

Reviewer's Responses to Questions

**Comments to the Author**

1. If the authors have adequately addressed your comments raised in a previous round of review and you feel that this manuscript is now acceptable for publication, you may indicate that here to bypass the “Comments to the Author” section, enter your conflict of interest statement in the “Confidential to Editor” section, and submit your "Accept" recommendation.

Reviewer #3: (No Response)

2. Is the manuscript technically sound, and do the data support the conclusions?

Reviewer #3: Partly

3. Has the statistical analysis been performed appropriately and rigorously? 

Reviewer #3: Yes

4. Have the authors made all data underlying the findings in their manuscript fully available?

Reviewer #3: Yes

5. Is the manuscript presented in an intelligible fashion and written in standard English?

Reviewer #3: No

6. Review Comments to the Author

Reviewer #3: From a study design standpoint, this is a relatively well conducted systematic review, but with some possible flaws. In my opinion, having decided to include only English language studies may have introduced a study selection bias as well as a publication bias. Why decide not to include also articles produced in the languages of the Arab member states of the GCC or other languages that can be spread in that geographical area?

I believe that a greater number of databases could have been explored (for example CINHAL and others of psychological and psychiatric interest such as PubPsych or PsyARTICLES and others).

Furthermore, it may have had repercussions on the heterogeneity and transferability of the results.

The possibility of introducing generic biases in the review was considered in the analysis of the results at line 265, but was not elaborated on in the discussion.

I believe it would have been more appropriate to use PRISMA in planning audit reporting and that this would also have been helpful in assessing the quality of the studies.

In general, the discussion of the results should be strengthened in relation to the results and potential limitations of the review itself.

Authors should try and

expand the meaning of their findings over and above the mere technical results.

The review is certainly useful in providing information, albeit not extensive, of the possible information gaps in the MHL area in GCC.

7. PLOS authors have the option to publish the peer review history of their article (what does this mean?). If published, this will include your full peer review and any attached files.

Reviewer #3: **Yes: **Silvia Pregno

---

## [Author Response · Author response to Decision Letter 1]

3 Oct 2020

Response to reviewers

General Response to Comments: 

We are appreciative of the raised comments and feedback by the Academic Editor and the reviewers, and made all possible efforts to respond to each of their comments. We have made minor revisions in the manuscript. These revisions are tracked and colored with red font. The line numbers mentioned in our responses below refer to those in the tracked version of the manuscript. 

ACADEMIC EDITOR: 

We suggest in particular you go through the PRISMA checklist providing explanations for all the items that were not met and include a synthesis in the limitations of the study. For example, you may wish to expand on the reasons why you only included articles in English and you did not consider publications in Arab Languages, as suggested by Reviewer 3. You may elaborate also on the choice to include a selected number of articles databases and how this may have restricted the number of publications considered in your review.

Response:

Thank you for your valued feedback, we have revised the PRISMA checklist and added more details providing explanations for all the items that were not met and included a synthesis in the limitations of the study. 

Please look at the following items in the PRIMSA checklist: 

14, 15, 16, 20, 21, 23, 25 

Please look at their corresponding pages’ numbers as mentioned in the checklist as well 

REVIEWERS COMMENTS: 

From a study design standpoint, this is a relatively well conducted systematic review, but with some possible flaws. In my opinion, having decided to include only English language studies may have introduced a study selection bias as well as a publication bias. Why decide not to include also articles produced in the languages of the Arab member states of the GCC or other languages that can be spread in that geographical area?

I believe that a greater number of databases could have been explored (for example CINHAL and others of psychological and psychiatric interest such as PubPsych or PsyARTICLES and others). Furthermore, it may have had repercussions on the heterogeneity and transferability of the results. The possibility of introducing generic biases in the review was considered in the analysis of the results at line 265, but was not elaborated on in the discussion.

I believe it would have been more appropriate to use PRISMA in planning audit reporting and that this would also have been helpful in assessing the quality of the studies.

In general, the discussion of the results should be strengthened in relation to the results and potential limitations of the review itself. Authors should try and expand the meaning of their findings over and above the mere technical results.

The review is certainly useful in providing information, albeit not extensive, of the possible information gaps in the MHL area in GCC.

Response:

Thank you for your insightful feedback. We have addressed your comments and the manuscript has been modified accordingly. Kindly see the details highlighted below:

Lines 119-124:

Additionally, for the results synthesis in this systematic review, we have critically reviewed and scrutinized the results and discussion of each study for outcomes, limitations and bias that authors may have had highlighted in their studies. Following that we planned to combine and examine various related ideas in literature, to show how included results, outcomes, and limitations fit together, and present them in a unified form. Finally, we used the items from CASP tool to draw limitations that were faced in this review, and further elaborate on them in the discussion.

Lines 202-206:

Furthermore, Quantifying the differences in means of levels of mental health literacy that were reported across included studies was not possible due to differences in measuring the outcomes and the scales used in addition to the selection of certain mental diseases to assess the level of mental health literacy related to them across different population groups. For example, adult females, health care providers or general adults.

Lines 260-261:

in the Arab region, where mostly both men and women tend to link emotional and mental issues with religious and social believes.

Lines 268-271:

This might potentially create a real barrier in achieving accessible and effective mental health services to the public and especially to vulnerable groups within the GCC region who may struggle to reach out for their primary health provider voluntary. 

Lines 281-297:

This review was limited to peer-reviewed articles published in English, which could result in bias. Countries in the GCC uses Arabic as the first language spoken, however, English is the second Language spoken and the formal language used in medical field including both practice as well as academic and scientific professional research. There, is only a few numbers database that covers articles written originally in Arabic language, but they mostly cover only limited numbers of articles from wide range of topic areas such as art and engineering, but not exclusively on health nor mental health in specific. Hence choosing English was the best choice that would yield more studies in searching. While extensive search was conducted, it is possible that relevant articles were not identified, as authors were using specific scientific databases. The three databases selected were among the most commonly used in systematic reviews. Relating to similar literature on the topic from the region, these same databases were commonly and most frequently used. Never the less, including more data bases could have yield more articles, but it wasn’t feasible nor convenient to add more search engine in the allocated time of the review. Authors also did not include information from other sources, such as unpublished reports from educational institutions or relevant literature. Our authors did not contact other authors to clarify vague information in reviewed studies. Thus, the evaluations may poorly assess the study quality, when details are not included in the reports

JOURNAL REQUIREMENT:

While revising your submission, please upload your figure files to the Preflight Analysis and Conversion Engine (PACE) digital diagnostic tool, https://pacev2.apexcovantage.com/. PACE helps ensure that figures meet PLOS requirements. 

 Response:

We have checked the style used in our manuscript against those required by PLOS

ONE using the recommended tools mentioned.

---

## [Decision Letter · Decision Letter 2]

23 Dec 2020

Mental health literacy in Arab states of the Gulf Cooperation Council: A systematic review

PONE-D-20-06289R2

Dear Dr. Elyamani,

We’re pleased to inform you that your manuscript has been judged scientifically suitable for publication and will be formally accepted for publication once it meets all outstanding technical requirements.

Kind regards,

Kyoung-Sae Na, M.D.

Academic Editor

PLOS ONE

Additional Editor Comments (optional):

Reviewers' comments:

Reviewer's Responses to Questions

**Comments to the Author**

1. If the authors have adequately addressed your comments raised in a previous round of review and you feel that this manuscript is now acceptable for publication, you may indicate that here to bypass the “Comments to the Author” section, enter your conflict of interest statement in the “Confidential to Editor” section, and submit your "Accept" recommendation.

Reviewer #3: All comments have been addressed

2. Is the manuscript technically sound, and do the data support the conclusions?

Reviewer #3: Yes

3. Has the statistical analysis been performed appropriately and rigorously? 

Reviewer #3: N/A

4. Have the authors made all data underlying the findings in their manuscript fully available?

Reviewer #3: Yes

5. Is the manuscript presented in an intelligible fashion and written in standard English?

Reviewer #3: Yes

6. Review Comments to the Author

Reviewer #3: Dear Authors, you have adequately addressed my comments raised in the previous round of review and I feel that this manuscript is now acceptable for publication and that it is offering deeper findings . Thank you.

7. PLOS authors have the option to publish the peer review history of their article (what does this mean?). If published, this will include your full peer review and any attached files.

Reviewer #3: **Yes: **Silvia Pregno

---

## [Editor Report · Acceptance letter]

28 Dec 2020

PONE-D-20-06289R2 

Mental health literacy in Arab states of the Gulf Cooperation Council: A systematic review 

Dear Dr. Elyamani:

I'm pleased to inform you that your manuscript has been deemed suitable for publication in PLOS ONE. Congratulations! Your manuscript is now with our production department. 

Kind regards, 

on behalf of

Dr. Kyoung-Sae Na 

Academic Editor

PLOS ONE